# Aligning Community-Engaged Research to Context

**DOI:** 10.3390/ijerph17041187

**Published:** 2020-02-13

**Authors:** Jonathan K. London, Krista A. Haapanen, Ann Backus, Savannah M. Mack, Marti Lindsey, Karen Andrade

**Affiliations:** 1UC Davis Department of Human Ecology, University of California, Davis, CA 95616, USA; 2Department of Human and Organizational Development, Vanderbilt University, Nashville, TN 37235, USA; krista.a.haapanen@vanderbilt.edu; 3Harvard Chan School of Public Health, Boston, MA 02115, USA; abackus@hsph.harvard.edu; 4UC Davis Center for Health and the Environment, University of California, Davis, CA 95616, USA; smack@ucdavis.edu; 5School of Pharmacy, University of Arizona, Tucson, AZ 85721, USA; lindsey@pharmacy.arizona.edu; 6Stanford University School of Medicine, Stanford, CA 94305, USA; kandrade@stanford.edu

**Keywords:** environmental health science, environmental justice, community based participatory research, community–university partnerships

## Abstract

Community-engaged research is understood as existing on a continuum from less to more community engagement, defined by participation and decision-making authority. It has been widely assumed that more is better than less engagement. However, we argue that what makes for good community engagement is not simply the extent but the fit or alignment between the intended approach and the various contexts shaping the research projects. This article draws on case studies from three Community Engagement Cores (CECs) of NIEHS-funded Environmental Health Science Core Centers (Harvard University, UC Davis and University of Arizona,) to illustrate the ways in which community engagement approaches have been fit to different contexts and the successes and challenges experienced in each case. We analyze the processes through which the CECs work with researchers and community leaders to develop place-based community engagement approaches and find that different strategies are called for to fit distinct contexts. We find that alignment of the scale and scope of the environmental health issue and related research project, the capacities and resources of the researchers and community leaders, and the influences of the sociopolitical environment are critical for understanding and designing effective and equitable engagement approaches. These cases demonstrate that the types and degrees of alignment in community-engaged research projects are dynamic and evolve over time. Based on this analysis, we recommend that CBPR scholars and practitioners select a range of project planning and management techniques for designing and implementing their collaborative research approaches and both expect and allow for the dynamic and changing nature of alignment.

## 1. Introduction

Research conducted in collaboration with non-academic stakeholders has seen a substantial increase in popularity in recent decades [1,2,3]. Through these collaborations, local partners gain access to scientific resources and knowledge that can help inform community advocacy and bolster legitimacy in political and other public discourse, as well as help shape the research agenda of universities to respond to community priorities [4]. Researchers gain the firsthand knowledge and insight of local partners, develop interventions with greater relevance and feasibility, build bridges between the university and broader community, and support the self-empowerment of disadvantaged communities to take part in the production and mobilization of knowledge. Participatory and community-engaged approaches to research have been employed in a range of fields including social work [5], education [6], development [7], public health [8], and ecology [9].

Due to the well-recognized benefits of community engagement, such approaches to research have been translated into funding initiatives by a range of private foundations and public organizations, as well as into policy statements by state and federal governments. The National Institute of Environmental Health Sciences (NIEHS) has demonstrated a strong commitment to promoting community engagement through its Environmental Health Sciences Core Centers, Superfund Research Program, and NIEHS/EPA Children’s Environmental Health and Disease Prevention Research Centers. Each of these programs includes support for a Community Engagement Core (CEC) or Community Outreach and Translation Core (COTC), which are designed to facilitate bidirectional interaction between community stakeholders and researchers and translation of research into action. While community engagement can include a wide range of activities, the NIEHS has recognized Community Based Participatory Research (CBPR)—one form of community-engaged research—as a particularly effective tool for understanding and addressing environmental health concerns [10].

Although definitions of CBPR draw from some of the same principles, no single definition has been accepted across the field of scholarship or practice. Perhaps the most widely recognized parameter used in defining CBPR is the extent to which the non-academic partners are involved in the research process. One foundation of this framework originates from planning scholar Sherry Arnstein’s (1969) ‘Ladder of Citizen Participation (While Arnstein’s original study used the term “citizen”, we use the term “community” to avoid the problem of limiting inclusion to citizens, especially in a time of anti-immigrant discourse and policies. At the same time, we also acknowledge the problems associated with the term “community” as masking divisions and hierarchies.) [11]. This ladder represents community participation in determining federal social programs as existing on a continuum from “nonparticipation” to “tokenism” and finally to “citizen power.” Her continuum highlighted a concept that remains central to our understanding of CBPR today: that community participation in a program or process must be assessed based on the degree to which participants have the real power to affect the outcome of that process. The involvement of non-academics in research has since been conceptualized as falling along a continuum similar to Arnstein’s, with “nonparticipation” corresponding to conventional academic research, “tokenism” referring to research partnerships in which no real power is vested in community partners, and “community power” occurring when community partners take the lead in determining the questions, methods, and applications of the research. This continuum concept, which promotes high levels of active involvement and authority of the non-academic partner, has been widely accepted among CBPR scholars. 

The idea that “more is better” in community engagement can be seen in a number of widely accepted definitions of CBPR. Israel et al. [8] (p. 178) defined CBPR using eight principles, including “facilitates collaborative partnerships in *all phases* of the research” (emphasis added). Their CPBR typology encourages very high levels of involvement by members of the affected community, starting with selection of the issue of concern and continuing throughout the development of the research question and design, data collection, dissemination of the results, and application of the findings to social action. Other scholars have proposed that only the projects exhibiting the highest levels of community involvement and authority qualify as CBPR [12,13].

Despite the popularity of these definitions of CBPR, other scholars employ the term to describe a much broader array of translation-focused, partnership approaches to research [1,14,15]. Indeed, Spears Johnson et al. [16] revealed that, of 25 CBPR health projects funded by the Centers for Disease Control and NIH, community involvement ranged from 2 out of 13 defined phases of the research project to all 13. While some studies reflected Israel and colleagues’ [8] definition of CBPR and demonstrated high levels of non-academic partner authority throughout the project, others involved community partners in only the participant recruitment or dissemination of the results phases. This inconsistency suggests that scholars are using the term CBPR to describe projects with very different approaches and levels community involvement. One way to interpret these findings is that some projects are being inappropriately classified as CBPR and in some cases this is likely correct (for example in projects that only use community organizations to recruit study participants). It is certainly clear that the level of community involvement required to classify a study as CBPR is inconsistently interpreted and represented in the literature. However, these findings also raise the question of whether a linear “less/more continuum” alone provides an adequate basis for studying or applying CBPR in a wide and changing variety of settings or whether other frameworks might be more suitable for describing and designing context-appropriate strategies.

This paper emerged from the authors’ search for a CBPR framework that could support our work facilitating community engagement among a diverse array of environmental health sciences researchers and community partners. In our roles as CEC leaders for NIEHS-funded research centers, we work closely with university and community partners to support community engagement activities. Our search for a framework that could explain the successes and challenges of our work revealed an abundance of definitions and approaches, many of which were too prescriptive to fit our diverse contexts. We also recognized that each of our projects demanded different CBPR approaches and that the level of our success (or challenge) was not dependent solely on the extent of community engagement but also how our approaches fit our specific contexts. We therefore sought to develop approaches to research partnerships that would more effectively capture the individual capacities, organizational or institutional settings, and objectives of our constituents.

Building from the large body of literature characterizing and promoting research involving communities described above as well as our own decades of experiences facilitating community engagement in the environmental health sciences and other contexts [4,17,18,19,20,21], we propose a framework for facilitating and sharing knowledge about community-engaged research that emphasizes alignment between *what* the project intends to accomplish and *how* it will be undertaken.

Recognizing the diversity of factors that can shape community engagement in a given research project, we suggest that “good” community engagement does not always *maximize* the extent of community involvement but instead *optimizes* that involvement based on the alignment between the collaborators’ interests, capacities, and sociopolitical contexts. It is crucial to ensure that, given historical power disparities between academic institutions on the one hand and low-income communities and communities of color on the other, that alignment does not privilege the academic partners’ goals over the non-academic partners. Instead, our approach to alignment in CBPR is based on meeting community needs and interests as its primary goal. This multidimensional framework highlights the crucial roles that CECs and similar organizations within the research institution play in developing and maintaining this alignment as well as the value of clear communication about the expectations, goals, and concerns of the range of collaborators.

Figure 1 depicts the authors’ framework for developing a well-aligned community engagement approach. There are two sets of converging circles to represent the fact that the partnerships are formed by intersecting relationships between the roles of the community and the university partners. Each set of concentric circles (scale/scope; capacities/resources; sociopolitical environment) in the diagram represents different levels of contexts and influences on a given CBPR project. All of these contexts must be taken into account in developing a well-aligned community engagement approach. The boxes in the community engagement approach represent the different factors (research methodology, collaborator roles, and implementation plan) that can be considered when creating alignment between the CBPR project and its context. This framework can be used as an assessment tool for CBPR projects, evaluating how well these research design factors align with the context (scale/scope; capacities/resources; sociopolitical environment) of a given project. They can also serve as a guide to help CBPR scholars and practitioners adapt to tensions and enhance their projects over time [19]. We now address how to assess the alignment of each element of context.

### 1.1. Scale and Scope

At the core of successful CBPR partnerships is the alignment of project goals for the researcher and the community. The *scale* of CBPR goals can vary considerably—from tribe, to neighborhood to state- and to nation-scale impacts—and the *scope* of goals can range from policy change to increases in health literacy to decreases in health disparities. Project partners must find a set of shared goals and deliverables in order for the project to be mutually beneficial. This does not mean that the goals of the researchers and the community need to be exactly the same. The goals are often a combination of very different but hopefully complementary goals by the partners (e.g., the researcher’s goal may be a manuscript, the community partners’ goal may be data to inform a policy or campaign). The scale and scope of partners’ goals may influence such elements as the type of data collected and the collection method [22], the timeline of the project [23] and the number and type of collaborators (academic and non-academic). Agreeing on shared goals and deliverables does not mean that one or both parties needs to compromise their core interests, but that an overarching framework is needed that can accommodate both sets, and ideally find as much common ground as possible. However, as noted above, the community-generated scope and scale of goals should take precedence in developing this overarching framework. Effective projects are those in which there is an explicit process to identify the scale and scope of goals and as well as the means to continually assess and adjust this alignment of goals over the course of the project.

### 1.2. Partnership Capacities and Resources

Even when partners have established alignment in their project goals, their respective capacities and resources—on an individual or organizational/institutional level—may act as facilitating or limiting factors to achieving these goals. The community engagement approach must be developed in a way that neither over- nor under-estimates the capacities of each collaborator, that values each collaborator’s time and expertise, and provides each collaborator with an appropriate level of financial compensation or other resources. If the community engagement approach does not align with the capacities and resources of the collaborators, projects may overburden or undercompensate one or more collaborators, hinder participation, and/or fail to achieve one or more collaborators’ goals. An assessment of capacities must be conducted with both the academic and non-academic partners, including approaches to build needed capacities (for both sets of partners) to develop a project that meets all parties’ needs and interests. Obtaining additional resources to address any shortfalls or disparities is often a necessary step.

### 1.3. Sociopolitical Environment

The sociopolitical environment of a CBPR project is shaped by power relations between social groups, dominant political attitudes, population demographics, and historical relationships between the university and communities [24,25]. These contexts can in turn affect the availability of research funding, the willingness of non-academic partners to engage with researchers, the willingness of researchers to conduct CBPR, and the project’s likelihood of achieving political or behavioral change [25]. These contexts can be experienced at multiple scales. At the community scale, it is important to recognize the variety of types of associations. These associations can range from community of place (e.g., residents of a neighborhood or region), a community of identity (e.g., defined by race, tribe, ethnicity, socioeconomic status), community of affiliation (e.g., members of a social movement, religious or occupational group), or other form [26,27]. Communities are also not monolithic and dynamics of conflict, collaboration and hierarchies must be addressed [28,29,30]. In sum, ensuring strong alignment with the sociopolitical environment requires both an astute assessment of local dynamics as well as processes to adapt the project to changing realities throughout the project. It is often the community partners who have the insider knowledge to provide these assessments and adaptation strategies.

### 1.4. CBPR Approach: Research Methodology, Collaborator Roles, Implementation Plan

To develop an effective and equitable partnership between the researchers and the community members, the combined team must develop a shared understanding of the scale/scope, partner capacity, and sociopolitical context. This shared understanding will then form the information basis to create a context-appropriate CBPR approach. This approach includes the processes for building and maintaining trusting relationships between all partners, designing a research methodology that answers the questions identified by the community partners, and designating substantive roles for each partner. The three cases below illustrate these elements of the CBPR approach.

Below is a three-by-three matrix the authors developed (Figure 2) to illustrate this approach. The matrix calls out the combined measure of the three factors of the *alignment of the project to its context* (i.e., scale and scope of project goals, institutional capacities, and the sociopolitical environment) and the *extent of the community participation*. In this figure, alignment measures the degree to which projects have agreement on the scale and scope of project goals, appropriate levels and kinds of project capacities and resources, and that accommodate sociopolitical dynamics affecting the project. The extent of participation refers to the quality and quantity of roles that community partners play in the project (e.g., problem definition, research design, data collection, analysis, documentation, application). The alignment of all three elements of context can be improved though early and consistent dialogue between partners, a careful eye for disparities, an openness to adjusting the program structure over time. The engagement is tracked over time for the extent and alignment of community engagement through dashed arrows. This accounts for the dynamism of such projects and emphasizes that engagement is a process that continues to evolve over the life of a project or across multiple projects in a given institution such as a CEC [19]. Figure 2 can be used as an assessment tool and guide for tracking progress and guiding improvements on CBPR projects over time.

## 2. Materials and Methods

This article uses a comparative case study method to examine three different manifestations of CBPR in university–community partnership contexts. The intention of the comparison is to construct a typology of CBPR partnerships by drawing out both the distinctions and commonalities of the cases. These cases are drawn from projects from within Environmental Health Sciences Core Centers (EHSCCs) funded by the National Institute of Environmental Health Sciences at Harvard University, University of California, Davis and the University of Arizona. The Community Engagement Cores in all three EHSCCs are committed to supporting research that is mutually beneficial, mutually respectful, and that promotes multi-directional learning between university researchers and community partners. They also all use a variety of methods to build the capacity of both university and community partners to engage in such productive partnerships. This variety of methods reflects the specific contexts and goals of the research partnerships. We selected the case studies to illustrate the diversity of community-engagement approaches used by the Centers. One aspect of this diversity is the role played by the CEC: while the UC Davis CEC primarily operates as a facilitator and steward for CBPR projects, the Harvard case demonstrates how CECs themselves can act as research partners in CBPR projects. The University of Arizona case illustrates the CEC’s approach in aggregate in order to protect the privacy of their tribal partners. In each case a matrix of community engagement (illustrated above) is provided to indicate the levels of engagement and the changes over time. These matrices will provide a way to compare the three cases and to show the dynamism of each of the cases. We also present summary tables for each case study. Throughout, we also call out challenges in each case and how the leaders of the CEC and community partners worked to address them.

We seek to answer the following research questions.
In what ways and to what extent did each project or initiative align with the scale and scope of its goals, with its capacities and resources, and with its sociopolitical environment?What were the processes involved in developing and implementing the CBPR approach?What were the implications of these approaches on the outputs and outcomes of the projects?

By answering these questions across all three cases, we hope to inform the development of a conceptual and assessment framework that can be used for those engaged in CBPR projects whether in academic or non-academic institutions.

We used an action-research methodology to construct the case studies. In particular, we drew our data from the critical reflections of the experiences of the co-authors in designing and implementing the CBPR projects as well as an extensive dialogue between the project leaders about the similar and divergent processes and outcomes of their work. Qualitative data were collected through field notes from the lead researchers developed during and following the projects. Using a comparative approach of projects with three different institutional, topical, and methodological contexts is useful for developing a generalizable analysis that a single case study could not accomplish.

## 3. Results and Discussion

### 3.1. Case Study 1: Exploring The Asthma/Obesity Connection in Response to Community Concerns (Harvard University)

#### 3.1.1. Project Context: Partners, Roles and Implementation Process

This case study follows a course of action which the Community Engagement Core (CEC) of the Harvard Chan-NIEHS Center for Environmental Health (Harvard Chan Center) undertook to address the health concerns expressed at a Dorchester (Massachusetts) neighborhood workshop sponsored by the Boston Public Health Commission. The Harvard Center includes researchers who study the impact of exposures related to particulate matter (The Harvard EHSC also focuses on research related to persistent organics and metals.). As in the case studies that follow this one, this case examines the alignment among the collaborators in terms of the project’s scale and scope, its capacities and resources, and the sociopolitical climate within the context of engagement.

Dorchester is a community located south of and adjacent to Boston. It is bordered by the Atlantic Ocean to the east and toward a collection of towns and cities in the other directions that display a rich history and strong traditions. It has an ethnically and socioeconomically diverse population and, like other settlements in the area, a high population density. Route I-93, an intensely used interstate highway that services Boston, runs along the eastern boundary of Dorchester and next to the Atlantic Ocean coastline. This highway is a source of particulate matter pollution.

In 2012, the Boston Public Health Commission (BPHC) held a series of neighborhood meetings to identify the health concerns of Boston communities. Ann Backus, a leader of the Harvard NIEHS CEC, was a facilitator at the Dorchester meeting. The BPHC introduced the meeting with health data from the neighborhood. Meeting participants pointed to high traffic volume as responsible for poor air quality, dense residential neighborhoods, and a lack of parks and green spaces in Dorchester compared to surrounding neighborhoods. Participants entreated the city to mark bicycle lanes for safety, provide a bicycle rental program, and increase the number of biking and walking paths in order to encourage more physical exercise. The consensus of the meeting was that asthma and obesity were/are major health concerns in Dorchester.

Past and present asthma and obesity data from Dorchester support the community’s recognition of these health issues. The emergency department visits of children aged 3 to 5 per 10,000 population in North and South Dorchester, exceeded that of neighboring Boston by 114 and 126, 3-5-year-old children per 10,000 population, respectively. The prevalence of adult asthma was 11.8% in Boston while it topped 18.1% in North Dorchester. Similar exceedances hold for obesity: 21.8% in Boston compared to 28.8% and 26.3% in North and South Dorchester, respectively [31].

After reviewing the research of Harvard Chan Center’s Dr. Stephanie Shore and her lab as well as research from Johns Hopkins [32], the Center submitted a CEC proposal as part of its P30 resubmission to NIEHS in 2013 to explore the intersection of asthma and obesity and develop interventions for at least one of the audiences: community at large or health care professionals. The P30 was funded for the period 2014−2019. Thus, the idea to explore the connection between asthma and obesity began in Dorchester in 2012 as a community-city-university partnership.

In preparation for delving into the intersection of asthma and obesity and developing interventions, the CEC undertook several activities to understand the community’s concerns about air pollution and obesity, separately. The CEC sponsored a Science Café on in which transportation surfaced as a major issue—both how traffic contributes to pollution and how unsafe the community feels that bicycling is in the congested city of Dorchester. Programs such as the Boston Complete Streets Program that focused on multimodal accessibility, green and smart streets, including traffic calming, inclusion of bike lanes, and improvement of walkability and accessibility, were underway, but the community wanted more ways to address asthma and obesity.

After scoping out community concerns and some interventions, the CEC proceeded to “pilot” some interventions of its own. In 2016, Traci Brown, a post-doctoral scholar with Dr. Shore’s lab and a volunteer, catapulted the asthma/obesity project into high gear, and the CEC began to develop a broader community involvement and infrastructure for the project. In the parlance of this article, the CEC was exploring the alignment, in terms of scope and scale, between its efforts to understand the asthma-obesity connection and the concerns of the community. The CEC was also testing out the capacity and resources of the Center for addressing these concerns.

Dr. Brown, an expert hula-hooper, developed a ‘whole health hooping’ after school activity at the Dorchester Boys and Girls Club that encouraged exercise and mindful eating. She also formalized the Harvard Chan Centers outreach activities at the Dorchester Winter Farmer’s Market into a ‘Science at the Market’ format which included peak flow measurements, LEGO™ activities to demonstrate particulate formation from incomplete combustion, and inflatable lungs—all designed to teach about airway and lung health and to provide an opportunity to discuss smoking, asthma, and air pollution. This phase helped develop more research capacity for understanding asthma and obesity and allowed the CEC to contribute its information resources to the community.

At this point, the project became formally known as the Asthma/Obesity Connection (A/O Connection). Three aims were defined: (1) to gather information about the A/O connection; (2) to identify a constituency that would find this information useful; and (3) to develop an intervention that would include development of materials and dissemination of information about the A/O connection and its implications for public health.

To achieve Aim 1, the CEC undertook a literature review, engaged with medical professionals treating asthma or obesity, and used Dr. Shore and her lab team as a guide to test the strength of the literature and empirical experience of medical experts with respect to the A/O connection. Thus, the researchers and medical experts became the “knowledge generator network.” In this network, the flow of information was bi-directional and iterative. A healthy bi-directional exchange of information, concepts, and presentations developed between the CEC staff, researchers, and medical experts.

To achieve Aim 2, the CEC established a relationship with the directors of the Community Health Worker Programs in the asthma and nutrition divisions of the Boston Public Health Commission. The directors provided feedback and suggestions regarding the information that had been gathered regarding the A/O Connection through the knowledge generator network, and the CHW programs became the CEC’s “practice/users network.”

In May 2018, the CEC hosted a workshop on the A/O Connection titled “Asthma and Obesity: Approaches to Care.” This workshop brought together the knowledge and the practice networks. The CEC intentionally increased the practice network to include staff from non-profit organizations and from community health centers in the Boston area in order to improve the alignment of scope and scale between the generators and users and more effectively inform Aim 3, the development and dissemination of materials.

The sociopolitical aspect of the engagement surfaced at this point when the community participants in this workshop pointed out that obesity has different meanings for different cultures; they cautioned the team about approaching obesity from a perspective of shame and encouraged them to consider adding-in the contribution of the built environment (lack of walking, biking and park or green spaces). For these and other reasons, the CEC focused their deliverable, a booklet for health professionals, broadly on “lifestyle factors” rather than narrowly on obesity. During the materials development phase of this work, the CEC Stakeholder Advisory Board, including Eugene Barros of the Boston Public Health Commission, Division of Healthy Homes and Dr. Alan Woolf, of the Pediatric Environmental Health Specialty Unit (PEHSU) at Boston Children’s Hospital helped inform and were heartily supportive of the work.

The final work product is a booklet for health care providers titled *Obesity and Asthma* that practitioners such as Community Health Workers, nurses, school nurses, and others can use as a resource with their clients and patients.

#### 3.1.2. Alignment of Project Scale and Scope

Initially, the community envisioned a community-scale project while the Center thought the project had the potential to reach beyond the community—perhaps to regional and national levels. Regarding scope, the community saw this project as having broad and complex scope in terms of local health issues; whereas the Center did not have an explicit idea about project scope. Thus, the community and Center were not well- aligned in terms of scale and scope.

The CEC began to correct this Center/community misalignment in the area of project scale and scope when the PhD toxicologist joined the CEC, undertook an extensive literature search and personally engaged with experts in the field, locally and nationally. She also developed strong relationships in the community with organizations that were either addressing asthma or obesity. These relationships netted the Center a cadre of practitioners who could be resources both to the Center during project development, and to the community in the future. Thus, by tightening the relationships with organizations that shared the goal of reducing asthma and obesity, the CEC began to trade misalignment for alignment with respect to the scale and scope of the project.

#### 3.1.3. Alignment of Partnership Capacity and Resources

In addition to low alignment of scale and scope in the early phases of the project, there was little alignment of capacity and resources. In spite of the fact that the Center has PM (particulate matter) as one of its areas of research, it had only one lab looking at airway health and obesity and then only in a limited way. The Center’s capacity to answer the question, “Is there a connection between asthma and obesity?” was slim. This meant that the Center did not have a cadre of researchers who could be pulled into the project on the asthma side.

On the obesity side, the Harvard Chan Nutrition Department was very ready to engage. A nationally known researcher in the area of nutrition and mindfulness, as well as others, gave generously of their time and expertise, and the CEC staff boosted their knowledge of nutrition and obesity by attending noontime nutrition seminars.

On the community side, the capacity was not ready-made either. Dr. Brown’s interviews with practicing asthma educators and physicians in the Boston area revealed that none had any notion about obesity as a risk factor for asthma, a concept that had become clear to the CEC as a result of the extensive literature review and the discussions with several practitioner/researchers in the fields of pulmonary and bariatric medicine. The divisions at Boston Public Health Commission that address asthma and obesity (nutrition, in general) were wholly separate entities, and their respective staffs had never met together on the asthma/obesity connection topic. In fact, BPHC would probably have continued to address asthma and obesity as separate concerns if the partners had not hypothesized that there might be a connection and intervened on that connection.

Once the CEC-based toxicologist had connected the CEC with the community and established a relationship with the asthma and nutrition staff at BPHC, the CEC then had direct access to the Community Health Workers who would be the primary users of the A/O materials. The capacity and resources of the community and of the Center increased markedly over the course of the project and came into strong alignment.

#### 3.1.4. Alignment with Sociopolitical Environment

In this project, the sociopolitical climate showed up in several contexts. First, perhaps it is worth mentioning that as far back as 1999, a research team at the Harvard School of Public Health consisting of renowned researchers, Dr. Walter Willett of the nutrition department and Dr. Frank Speizer of Environmental Health, had published an article titled, “Prospective study of body mass index, weight change and risk of adult-onset asthma in women” [33]. However, this article had no meaningful traction at the time with practitioners and was not introduced to the community. Therefore, when the community of Dorchester cited asthma and obesity as major health concerns in 2012, they were not referring to the asthma-obesity connection, but to asthma and obesity as separate health concerns. It was not until CEC Director Ann Backus proposed to explore the link between these two diseases that the ideas surfaced by Willett and Speizer in 1999 [33] were brought back into open discourse.

On the community side of the sociopolitical context, there was a misalignment in terms of cultural literacy. This misalignment was brought to the fore when the 2018 workshop participants specified that obesity has different meanings and values in different cultures. For example, whereas in Anglo cultures slimness may be desirable, in some African and African-American and Hispanic cultures a fuller body may be preferred. In this sociopolitical domain, the partners also discussed that obesity is often presented and/or understood in the context of shame, and the community cautioned the CEC that their materials should avoid a shaming or judgmental approach. This conversation was informative for the Center and improved its cultural literacy around obesity. Thus, in part to de-emphasize the strict relationship to obesity and give practitioners a more generic handle with which to talk with their clients and patients, the CEC chose to express the obesity intervention in terms of healthy lifestyle choices rather than weight loss. This approach also gave the CEC an opportunity to suggest that communities have a role to play with respect to advocating for infrastructure change (e.g., safe biking and walking routes, playgrounds and parks, rental bikes, marked roadways, and mitigating food deserts).

The historical inequities of lack of accessibility to health care and low-income housing were being addressed by the organizations with which the CEC had built relationships such as the Codman Square Community Health Center. Food desserts and lack of fresh produce were being addressed by Mattapan Food and Fitness Coalition and by the Winter Farmers’ Market (in which Harvard participated) sponsored by the Codman Square Community Health Center. Healthworks Community Fitness for women and children at the Codman Center offered and continues to offer low cost fitness programs and free childcare as well as The Daily Table—a low cost “grab-and-go” food shop—and a Teaching Kitchen. With the CEC’s relationships to Codman Square Community Health Center and Mattapan Food and Fitness, it was aligned with the socio-economic climate and perhaps by inference, the political climate as well.

The CEC is in the dissemination and adoption phase of this project. The *Obesity and Asthma Booklet for Health Care Providers* is supported by the Boston Public Health Commission and is available to the Community Health Workers in the asthma and nutrition divisions. The CEC expects to reach out to the BPHC Community Health Workers, again, and to the health care professionals and the various community health centers over the next months.

#### 3.1.5. Synthesis: Alignment of Scale and Scope, Capacities, and Sociopolitical Environment

Within this relatively narrow project vis a vis the others cited in this paper, the Asthma-Obesity Connection project went through different phases of alignment, starting with low alignment (of scale and scope) and low participation (a reflection on the limited capacity and resources). The movement of this project from low alignment to high alignment is the result of the following set of interventions to improve the capacity and resources of the partners and the partnership as a whole (Figure 3). The alignment in scope and scale was improved through the identification of a clear research question: (i.e., what/ is there a connection between asthma and obesity?) and the broadening of the research methodology to include literature review, bench science, plus interviews with experts and practitioner. The alignment in capacities and resources was improved through the work of a highly trained and motivated project coordinator, identification of a broad spectrum of collaborators; and the commitment of a well-positioned core local partner. Finally, the alignment with sociopolitical context was improved through the creation of a community of practice linking all partners to address local issues.

The project was largely initiated and supported by the CEC, a university entity. With that said, the concern for the A/O connection emerged directly from neighborhood meetings, and the project has been shaped by significant contributions from local policy leaders, health practitioners and neighborhood residents. In the matrix shown below, the change in alignment and participation over the course of the project is indicated by the arrows from low alignment/low participation to medium alignment/medium participation and then to high alignment/medium participation. The goal is of course to achieve both high alignment and high participation. Based on an assessment of the project to date, current steps to further enhance the alignment of and participation in the project are to obtain feedback from current users of the program, i.e., the community health workers in Boston, and reach out more broadly to asthma educators, pediatricians and other practitioners in the area with an updated version of the booklet based on feedback and new research.

### 3.2. Case Study 2: Building Partnerships for Air Quality Research: Imperial County PM Study (UC Davis)

#### 3.2.1. Project Context: Partners, Roles and Implementation Process

Imperial County is a largely agricultural and desert area in the southeast corner of California with the Salton Sea to the north and the US-Mexico border to the south. It is the site of several fierce environmental justice struggles over issues such as agricultural practices (e.g., concentrated animal feeding operations and field burning), drinking water contamination, cross-border air pollution, the heavily contaminated New River and the fate of the Salton Sea. The political climate is generally conservative and dominated by major economic interests while the majority Latino population contends with problems of poverty, political representation, and harsh occupational and environmental conditions. At the same time, these communities are also engaged in growing empowerment and organizing efforts.

The Imperial Valley has consistently ranked as the California county with the highest asthma rates in children [34,35]. Data from many studies, including the southern California six cities study, show that respiratory health problems in children can be worsened by increased levels of particle pollution [36]. Community advocates, often led by the Comité Civico del Valle (CCV), have worked to confront these challenges. CCV is the only environmental justice organization in the Imperial Valley, and thus takes on a wide range of issues and campaigns. CCV has significant experience in community-based air quality monitoring through collaborations with public agencies and university researchers as well as provision of technical assistance to other community-based monitoring networks throughout the state [37]. Leaders with CCV realized that creating an effective air quality strategy would require more evidence of a causal relationship between Imperial Valley’s elevated levels of particulate pollution and asthma rates.

The CEC of the UC Davis UC Davis Environmental Health Sciences Center (EHSC)—established in 2014—has a mission to connect community organizations with UC Davis researchers with relevant expertise to pursue collaborative research projects. As part of its annual call for pilot project studies, the CEC (then managed by post-doctoral scholar Karen Andrade and directed by professor Jonathan London) worked with Colin Bailey, then the Executive Director of the Environmental Justice Coalition for Water (EJCW) and member of the EHSC’s Community Stakeholder Advisory Committee and Luis Olmedo, director of CCV to develop a proposal. London had worked with both EJCW and CCV on earlier environmental justice research projects and Andrade had significant experience with CBPR. Given EJCW and CCV’s interest in air pollution and asthma, the CEC sought out EHSC-affiliated faculty with relevant expertise and identified Kent Pinkerton and his lab, which is internationally renowned for its expertise in the respiratory health effects of air pollutants [38,39,40,41,42]. The Pinkerton Lab, CCV, and EJCW received funding through the EHSC in the form of a two-year pilot project grant from the UC Davis EHSC with 15% going to CCV to support their active engagement in the project using a CBPR approach. Shortly after the project was funded, the CEC’s leadership team underwent a transition, and Karen Andrade left for another position in her field.

#### 3.2.2. Alignment of Project Scale and Scope

The project’s scientific goal was to differentiate the sources of PM in Imperial Valley and measure source-oriented PM toxicity and its potential to invoke or exacerbate an asthmatic response in a model. The proposal included an open-ended community action goal, which stated: “The results will be shared with the community so they can inform future action if they desire to do so. Future action may be translating and leveraging the results for policy advocacy or co-writing articles and funding proposals for further research.” This was an important outcome goal, but did not specify the roles of UC Davis or CCV in the research design and implementation itself.

The first meeting between the project collaborators included CCV’s CEO, Luis Olmedo, and lead air monitoring program coordinator Humberto Lugo; UC Davis air quality engineer, Keith Bein; the UC Davis Western Center for Agricultural Health and Safety’s (WCAHS) Education and Outreach Specialist, Teresa Andrews; and a PhD student, Savannah Mack. This comprised the UC Davis team which would be led primarily by Savannah Mack.

The UC Davis team described their proposed scientific plan and the group then discussed CCV’s previous research collaborations as a context for the upcoming project. CCV emphasized that they needed this to be a true collaboration that would have a lasting community impact, as opposed to many other collaborations that had ended with scientists collecting data and leaving. The UC Davis team was in strong agreement with this value. However, when the discussion turned to how this could be done with this project, it became apparent that both UC Davis and CCV expected the other party to develop the community engagement strategy and action plan. They ultimately agreed upon “learning more about PM in Imperial Valley” as a desired outcome. While inclusive of partner interests, this general goal would prove to provide insufficient guidance for the project.

#### 3.2.3. Alignment of Partnership Capacities and Resources

While the Pinkerton lab research team itself had no prior experience with CBPR, they did have the benefit of the Western Center’s Education and Outreach Specialist and the support of the CEC. They also had significant expertise in the scientific methods to be used in the project. In addition, CCV had extensive experience working with universities on other air quality monitoring projects. A key challenge of capacity that was difficult to overcome resulted from the ambiguity of roles and responsibilities for applying the research. Both teams lacked a clear vision of each other’s capacities and the training needed to bridge that gap. CCV was expecting UC Davis to have a plan to implement their results in the community, whereas UC Davis was expecting that guidance to come from CCV as the primary community partner.

Based on an overestimation of the strength of the collaborative relationships and the capacities of the UC Davis team, the CEC did not get directly involved in the early meetings. In hindsight, this direct engagement would have been very helpful. Several times throughout the course of the project when miscommunications between the UC Davis research team occurred, the CEC stepped in to moderate. This assistance in problem solving was very useful as the project navigated applying for new funding and made decisions on how to move the study forward. As the project evolved, however, the team realized that instead of a background coaching role, it would have been better if the CEC provided additional training in CBPR and had helped facilitate from the onset of project development and subsequent meetings.

To increase the level of community engagement, CCV recommended the creation of a Community Advisory Committee, as had been typical with their other CBPR projects, and one of their staff members took the initiative to organize a group of community members to participate. This group included community members from a variety of occupations and locations throughout Imperial Valley, all of whom had previously participated, or currently were participating in multiple advisory committees for CCV and therefore had strong background knowledge and valuable lived experience in the health and pollution problems in the region. After the creation and first meeting of this group (hosted by CCV but run by Savannah Mack), the Community Advisory Committee and CCV became increasingly integrated into the project. Although they did not participate in the sample collection or data analysis (this was conducted solely by the UC Davis researchers), the community partners’ input was quite valuable in developing the study design and considering applications of the data. For example, the advisory committee clearly identified that the Salton Sea was their main concern so the UC Davis researchers located the sampling site as close to that PM source as possible. With the help of the CCV’s Humberto Lugo, the UC Davis team identified and secured a sampling location on the grounds of a local high school. The Pinkerton Lab decided that, since sampling would occur on school grounds, close collaboration with the high school itself would be helpful in conducting remote sampling.

While placing the sampling unit, the UC Davis team formed a relationship with one of the high school teachers, and enlisted her students’ help to maintain the sampling unit between the researchers’ visits. Mack trained the high school students on the operation of the unit and engaged them in the data analysis as well. Mack then secured separate funding, in addition to the EHSC pilot grant, from the WCAHS, and established a week-long internship for high school students on the UC Davis campus. This internship continued for two summers and included students from the sampling-site-school as well as another high school in Imperial Valley.

The results of the air sampling came at a much slower pace than originally expected due to difficulty with the setup of the sampling trailer, and so by the time the project funding period had ended, no significant scientific conclusions could be made. The difference in timeline and lack of progress in this aspect of the project was disappointing to both CCV and to the UC Davis team, as was the fact that there was no health action plan to do something with the eventual results. Because of this, it was agreed that in the absence of data, no formal action plan could be produced.

In the year after the pilot ended, the UC Davis team secured additional funding from WCAHS to continue maintaining the sampling site, as well as to continue the high school internship for a second year. The research team collected and analyzed samples for two agricultural seasons and observed significant differences in the toxicity of particles based on size. As of this writing, Savannah Mack is analyzing particle size comparison data, as well as completing a seasonal comparison. The UC Davis researchers shared the results to date with the high school students during their internship and made plans for the students to share their new knowledge with their schools, respective communities as well as the advisory committee and CCV.

#### 3.2.4. Alignment with Sociopolitical Environment

As noted in the introduction, the sociopolitical environment of the region is marked by historical conflicts between community environmental justice and health equity organizations and the dominant agricultural interests with their allies in local and regional government. This puts significant pressure on community organizations such as CCV to confront these problems. In keeping with discussions with CCV staff, the UC Davis team intended to use the scientific results from this project to aid them in the funding proposals and advocacy campaigns to mitigate PM in the Imperial Valley. The UC Davis researchers attempted to align their work to address these challenging sociopolitical factors by designing a project that would provide information to inform CCV’s air quality action strategies. However, the lack of the final results and a public health action plan did not allow for the fulfillment of this goal. On the other hand, it did build human capital through the development of the student CBPR process and summer internship program for under-served youth.

#### 3.2.5. Synthesis: Alignment of Goals, Capacities, and Sociopolitical Environment

Assessing the overall alignment of the project with the Imperial Valley community is complex, partially because of the range of entities that make up the “community,” and the transformations of the partnerships over time.

The alignment of goals between CCV and UC Davis was moderate at the project’s inception as the UC Davis CEC and EJCW, with assistance from CCV, collaborated to write the funding proposal. This could have been higher if CCV had been more directly engaged in the project development. In particular, had CCV been more involved in this stage, their expectation that the university team develop a public health action plan may have been made explicit and subsequently integrated into the structure of the project.

Because CCV was engaged in several academic collaborations at the same time as their project with UC Davis, the distinction between the projects being conducted by various institutions in the region were not as clear as they could have been. This issue was amplified by the fact that UC Davis and CCV each had several team members involved in this project. Although Savannah Mack and Humberto Lugo from CCV, were working closely, the team did not sufficiently engage the CCV leadership. When Lugo moved to a different organization, the UC Davis team realized that the full scope of activities and connections developed with him had not been integrated into the organization as a whole.

The alignment between the UC Davis team and the Community Advisory Committee was strong throughout the project as it served as a forum to identify local needs and interests in the project and to shape the study design to meet them. One advisory committee member observed that this project was “necessary and important.” One institutional challenge was that, while the Community Advisory Committee had been developed in collaboration with CCV, it was facilitated solely by the UC Davis research team and therefore did not have a close enough connection to build co-ownership with CCV.

Finally, there was very strong alignment between the goals and the activities of the UC Davis research team and the teachers and students at the local high schools. The educational value of the project as well as the internship was mutually appreciated. The principal from Calipatria High School said, “The partnership has a tremendous effect on the students.” One student evaluation said: “I want to share what we learned with our community and feel like I now have the skills to do so.” Like the Community Advisory Committee, however, because this element of the project was not directly connected to the CCV’s own programming, it was not perceived by CCV as representing the sufficient community partnership that the UC Davis team had intended.

While challenged in many ways, the project has had multiple benefits. First, it has ignited a passion in Savannah Mack to complement her scientific expertise in toxicology with a CBPR approach. She intends to pursue a postdoc with a focus in CBPR and plans to create a program to help graduate students learn how to best implement CBPR in their scientific research. The student internship has also catalyzed the interests of local youth in environmental health science, as evidenced by their excitement to return for a second summer and the fact that several students are pursuing bachelor’s degrees in Environmental Justice. Additionally, the local teachers are integrating this project and air quality issues into their curriculum. The results themselves are promising in their ability to identify the sources and health consequences of air pollution in Imperial Valley and will be shared with CCV to help inform their air quality advocacy.

Finally, the assessment of the project through the alignment framework prompted a number of improvements in the CEC’s approach to the CBPR projects. First, the CEC now provides a robust training in CBPR principles and practices for all Pilot Project grantees at the beginning of their projects. It also has formalized the provision of funding (up to $20,000 out of a $60,000 project) for community partners. The CEC also works with the grantees and community partners to develop a MOU that spells out expectations, roles, and responsibilities. Furthermore, it now plays a more active role in facilitating project meetings, especially in the initial relationship-building phases and provides more intensive coaching on CBPR to researchers. Finally, it has developed a results report-back framework that includes community workshops, info-graphics and research briefs for community audiences.

The following figure (Figure 4) summarizes the extent and alignment of the community engagement throughout the life of the project. It includes three different trajectories for each of the three major stakeholders in the project: CCV, the Community Advisory Committee, and the students and teacher at the local school. The partnership with CCV began with mid-alignment and mid-participation and shifted to a lower level of alignment with the inability of the project to develop a public health action plan. The partnership with the Community Advisory Committee began with mid-alignment and high participation and shifted to high alignment and mid-participation. As the project ended its information and data collection phase, the advisory board was less involved, but with regular updates from the UC Davis team, Community Advisory Committee members showed continuous and increasing appreciation and support for the project, in particular the collaboration with local students. Finally, the partnership with the school began with high alignment and mid-participation and increased over time to a high degree of both, as students and the teachers became more involved and empowered through active roles in the research process.

### 3.3. Case Study 3. Preparing and Supporting Faculty for Community Engaged Research with Tribes: (University of Arizona)

#### 3.3.1. Project Context: Partners, Roles and Implementation Process

The CEC of the Southwest Environmental Health Sciences Center (SWEHSC) at the University of Arizona (UA) is working with researchers and tribal communities by seeking alignment within their joint projects of scale and scope, capacities, resources, and sociopolitical context. The SWEHSC is a collaborative and interdisciplinary research center, which is actively investigating the health effects of environmental factors in the desert southwest and serving as a resource for general and tribal communities, primarily in Arizona.

About one-quarter of Arizona is made up of American Indian communities, largely in rural areas of the state, serving as home to twenty-two federally recognized tribes (While the formal term is American Indian, other terms such as Native American, Indigenous, and Tribal are used interchangeably by these populations and this practice is followed in this case study). The majority of the members of the Navajo Nation, the largest American Indian nation in the United States, and of the Tohono O’odham Nation, the second largest, are located in Arizona. The state has the third highest number of American Indians of any state in the Union. Approximately 286,680 were estimated to live in Arizona, representing more than 10% of the country’s total Native American population of 2,752,158.

Arizona has an arid, beautiful and fragile natural environment, compounded by issues of water shortage and distribution. Residents of Arizona are exposed to environmental chemicals, naturally occurring metals, airborne particulate matter, and organic solvents produced by activities such as agriculture, building and road construction, and mining. Heat waves of extraordinarily high summertime temperatures for extended periods of time are becoming more common. Smoke from fires and dust from dry arid landscape have raised particulate matter (PM) concentrations. Investigators of the SWEHSC are conducting research to determine how these environmental factors are related to such diseases as chronic pulmonary disease and other types of lung problems, skin and other cancers, reproductive problems, and liver and kidney disorders, among others.

#### 3.3.2. Alignment of Project Scale and Scope

The CEC’s involvement with tribal communities began in 2003 with an Air Toxics Project in the Phoenix, AZ area, which included three tribes in close proximity to Phoenix. The project resulted in positive relationships, trust for the SWEHSC, and new materials about lung health and environmental exposures. A US EPA-funded Community Action for a Renewed Environment (CARE) project with one of the communities led to sixteen years of regular involvement of the CEC with the Inter Tribal Council of Arizona (ITCA) and environmental professionals in the tribes in Arizona and relationships with tribal members and leaders.

The CEC has hosted three Tribal Forums about the environment and human health in collaboration with tribal partners, regularly participated in Earth Day events in six communities, and provided environmental health information upon request about arsenic exposures, perchlorate, forest fires, air pollution and asthma. CEC staff and interns support tribal summer camps and other youth activities. In addition, the CEC and SWEHSC members are supporting two research projects based on tribal community questions about environmental public health around air pollution and health and one about educational principles for working with tribal transfer students.

#### 3.3.3. Partnership Capacities and Resources

The Air Toxics Project and the air quality and education citizen science projects represent a strong alignment between the SWEHSC and area tribes. This alignment is made possible through trusted relationships with individual professionals who represent the interests of tribal nations and with community members actively involved with tribal environmental, educational, and health departments.

However, there is also a legacy of misalignment related to the deep-seated distrust of research in tribal communities in Arizona because of past abuse and neglect by university researchers. In the Havasupai case, researchers at Arizona State University collected blood samples from the people to study genetic links to diabetes and used the samples for other studies, thus violating ethical human subjects research [43]. A more recent example was a medical educational researcher divulging tribal data concerning low academic achievement at a conference presentation without proper permission from the associated tribe. As a consequence, Arizona universities [44] created data sharing, tribal consultation and research guidance policies. To address this general distrust of universities, the SWEHSC CEC has taken a collaborative approach that has led to researchers being invited to communities and to community-led research in collaboration with researchers. Three recent research projects mentioned above have provided the opportunity for capacity-building among tribal professionals, students, and community members. The agreements with each tribal entity preclude providing more details in this article as the projects are all in early stages and the data regarding these projects belongs to the tribes and has not been approved for dissemination.

SWEHSC members have been actively involved with the research projects identified above at different levels, which has led to capacity-building for researchers and their students. Researchers and their students have taken air samples, in another case researchers have provided advice concerning using the International Study of Asthma and Allergies in Childhood (ISAAC) [45] approach to assessing the prevalence of asthma in the community, and in all cases researchers and their students are participating in science café presentations in the communities. The continuum of involvement includes: passive and active involvement and direct engagement.
“Passive involvement” such as allowing graduate students and postdocs to participate in tribal community engagement, informational materials review, co-development of materials, educational activities and social media posts based on their research programs. For example, students from one lab assisted the CEC at an Earth Day event by learning and delivering lessons from NIH curriculum “Chemicals, the Environment, and You” [46]. They reached over one hundred Tohono O’odham students from five grades.“Active involvement” by researchers and lab staff, such as materials development on subjects requested by community partners, presentations to youth groups, and providing research internships to native students. One example of this was providing internships to six students from the Navajo Nation as they completed the KEYS High School Student Research Internship [47]. Researchers and lab students provide their time, talent, and treasure to the community engagement enterprise, providing real world supervision experience. Researchers express that such students reawaken their enthusiasm for the research initiative.“Direct engagement” with tribal members by researchers, such as science café discussions on topics requested by communities (and not necessarily their current research program), presentations and participation in tribal forums on health and the environment, providing advice and input to citizen science projects, and active partnerships with research program development with tribal communities based on the tribe’s research questions. A skin cancer researcher provided an introductory talk about how environmental hazards enter the body and subsequently cause cancer, providing basic information and answering community members’ questions. The community members who attend science cafés report that they now see scientists as approachable people who have important information to share.

This continuum of involvement was developed to include as many researchers as possible who have different capacities and abilities to translate their research to the public. This is important because SWEHSC membership includes bench and mechanistic researchers as well as public health researchers.

#### 3.3.4. Alignment of Partnership Capacities and Resources

In order to understand the research of members, the CEC conducts three types of activities: interviews and meetings with researchers and their lab members and discussions with high school student interns who are assigned to those labs. The first approach involves interviewing researchers periodically to develop an understanding of their research and their career development. The CEC writes profiles about the researchers that include short bios and plain language descriptions of their research. These profiles also include suggestions for articles to read. The second approach is for the CEC to meet with the entire lab group to learn more about the research and to teach grad students and researchers to speak and write about their research in plain language. The third approach is to discuss the research with high school students placed into laboratory internships. These students communicate about their experiences in abstracts, posters and presentations, which helps the CEC gain a better understanding of the research, both the foundational understanding and the current cutting-edge research.

The CEC comes to understand the capacities and interests of the SWEHSC researchers and from that understanding designs capacity building, suggesting how to present to youth and community audiences and how to conduct science cafés in the SWEHSC model. In these presentations, they give a short presentation which includes a little personal or family information, their career path, and a little about their research in plain language. The researcher and community members then have a conversation about the general topic based on a questions and answers approach. This approach builds the capacity of community members to engage with and understand environmental health research, which both community members and researchers enjoy.

Another approach to capacity-building for community members is to have researchers, grad students and postdocs review or co-create public information materials. The process the CEC uses is to understand information generated by research and then to transform that information into plain language, which creates materials that are easy to read with short sentences and no words of more than two syllables [48]. The goal is to meet the principles of good environmental health literacy promotion by educating scientists to provide accurate information that is consumable by the general public with only a high school education. This approach engenders bi-directional trust.

#### 3.3.5. Aligning with the Sociopolitical Environment

The most important sociopolitical issues that the CEC must address revolve around tribal sovereignty. This means that each tribe is a separate government entity with individual cultural practices and organizational structures. The SWEHSC also operates under Arizona Board of Regents Tribal Consultation Policy & UA Guidelines for relationships with tribes.

The CEC developed and uses an Indigenous Research Framework (see below) [49] as a tool for program development and to help researchers understand important aspects to consider when engaging in collaborative projects with Indigenous communities. The framework identifies common themes within Indigenous communities that might be relevant and adaptable to individual tribal circumstances. It is comprised of three rings of influence for Indigenous communities. The outer ring represents the aspects of communities that researchers should understand and be respectful of when engaging in projects. The second ring represents the inherent sovereignty of Indigenous communities that researchers should respect. The center of the framework includes the four key components of collaborative projects with Indigenous communities: relationship-building, project planning, project execution, and evaluation. These four concepts follow a continuous circular pattern to represent that the process is never-ending, and that steps are constantly cycling throughout the course of a project.

#### 3.3.6. Synthesis and Lessons Learned: Alignment of Goals, Capacities and Sociopolitical Context

By following the above approaches, the SWEHSC’s CEC has increased both the extent of the community partner participation and the alignment of this work with the overall context (including partner goals, capacities and sociopolitical context). This is represented in Figure 5, which illustrates changes in the alignment in participation between 2003 and 2019.

Using the Indigenous Inquiry Framework is an essential guide for researchers and program staff as they carry out collaborations and partnerships with tribal communities. All community collaborations require excellent communication, commitment, and consciousness of community politics. However, relationships with tribal communities need to place those qualities within the context of tribal sovereignty and self-governance. The characteristics listed below [50] have proven useful for long-term productive relationships

Active listening to tribal partners means that it is essential to begin with listening and not talking in meetings, thinking and taking a pause before responding, and thinking if a response is even required. Respect for the partners’ abilities and point of view is essential as well as realizing that tribal communities have been successful in sustaining themselves for many generations and therefore know more than the academics about their needs and interests. Academics need to be present in the community frequently in order to build trust and need to interact as learners as much as—or more than—they do as teachers. On a logistical level, transportation plans need to be in place to sustain a relationship with monthly or more frequent meetings and in-person engagement. This is expensive in terms of the time and travel costs but cannot be minimized. Virtual contact methods are not equivalent replacements, but they can be a supplement.

Longevity and consistency are vital. Developing relationships takes years—not weeks or months—and cannot be rushed. Researchers should expect and plan to stay engaged long after a specific grant for best results. Best practices include meeting with tribes at least a year before grant submissions, developing follow-up projects and continuing to attend meetings after the grant concludes. Developing relationships with multiple contacts within the tribes is important because there are frequent staff changes and without such contacts the team’s intentions will have to be reestablished or risk being frozen out.

Understanding tribal systems and chain of command is crucial. Tribes are independent government bodies with elected leaders who should be approached first to consider resolutions to support the engagement program. Memoranda of Agreement (MOAs) will need to be established and renegotiated to reflect the changing nature of the work. MOAs are essential to ensure researchers are meeting tribal priorities and can continue to ensure mutual benefit.

Publishing should not be a primary goal and in some cases publishing may not be possible. To be successful with publications, equal authorship must be offered; the benefit of the project to the tribe must be clearly stated. Plans for publication need to be made at the beginning and not assumed. The tribal partner and the researcher or university staff members need to clearly understand what data will be collected and reported.

It is good practice to engage tribal students in the studies and to provide stipends to tribal members who participate. Overall, projects need to collect, analyze, and report data in culturally appropriate ways, as defined by the partner and their community. For example, education projects as defined by “Himdag” the Tohono O’odham cultural practices [51] can be a source of strength; such projects can serve as a protective factor contributing to wellness which is aligned with Tohono O’odham culture.

The following Figure 6 illustrates the journey of alignment of the University of Arizona’s CEC with local tribes. It uses a curved line to indicate while the partnerships generally moved towards high alignment and high extent of participation, there was significant variation across the multiple projects and initiatives presented in the case study.

## 4. Conclusions

The three case studies presented above illustrate a number of important lessons about the design and implementation of CBPR projects. While partners in all three projects had common overarching goals—the building of mutually-respectful and beneficial relationships, community empowerment and benefit, and a strong research to action translation—they approached the projects in distinct ways that affected the outcomes and impacts of the work.

In short, taken together, the three case studies demonstrate that the success of CBPR projects is based on several key components. At the broadest level, success is shaped not only by the extent of participation, but by the alignment of the three levels of context: scope/scale, capacities and resources, and sociopolitical environment. Because all projects exist in and are influenced by different contexts, there is not a one-size-fits-all model of CBPR that can achieve its intended outcomes. Second, the degree of success of CBPR projects/initiatives changes over time in response to changing internal and external circumstances. Adaptability to these changes is key to progress and success. Finally, a focus on building-relationships in the early stages of the project can help ensure success in the longer term. This allows for a collaborative development of the scope and scale of the project, assessment of the capacities/ resources and sociopolitical environment and adjustments of the project to fit these contexts. The three types of alignment (scope and scale; resources and capacities; and sociopolitical context) are summarized in Figure 7 below. 

In the case of Harvard University’s community partnerships on the asthma/ obesity connection, the scale and scope of the project’s goals did not start off as having all of the relevant partner capacities and resources but became increasingly well-aligned as new partners were added with the requisite types of expertise and networks. The project also brought together partners who were working on different parts of the issue (asthma and obesity) but who had not worked in a collaborative way on the interconnections between these issues. This allowed for a greater alignment with the sociopolitical environment and furthermore, helped shape this environment by translating research into health care protocols to address the asthma and obesity connection.

The UC Davis asthma and air quality study selected a topic that was very well-aligned with the sociopolitical environment of the Imperial Valley where such concerns were of great interest to local partners. The project also was well-aligned with the technical expertise of the research team and the community-engagement capacities of the main community partner. However, there was less of a connection between the partners on their emphasis on translating research to action, with both the partners expecting the other one to take on this element of the project. On the other hand, the project developed a very successful youth-engagement process with the local high schools, where the STEM pathway focus of the schools and the mentoring capacity of the researchers was closely matched. These experiences of challenge and success have led to important improvements in the programs of the UC Davis CEC.

Finally, the case of the tribal partnerships with the University of Arizona’s CEC highlights the value of focusing on university–community relationship building with close attention to both the fraught histories of such partnerships (in Arizona and elsewhere) and the specific contexts of Tribal Sovereignty and governance. The CEC’s emphasis on building the capacity of university researchers to engage respectfully and effectively with tribal residents and the development of community-based science communications programs both enhanced trust and facilitated the development of new environmental health projects and the broader application of existing ones. The provision of a continuum of engagement model (from passive to active to sustained engagement) allowed for options that fit the scope and scale of a range of different projects. Overall, the CEC’s strategy of building an on-going partnership that supported a diversity of projects allowed for a continued improvement of both extent and alignment of projects over time.

Together, these case studies offer, not a blueprint, but a framework, criteria for assessment and a menu of options that those interested in building community–university partnerships can draw from as they seek models that align their internal and institutional capacities with their external contexts.

## Figures and Tables

**Figure 1 ijerph-17-01187-f001:**
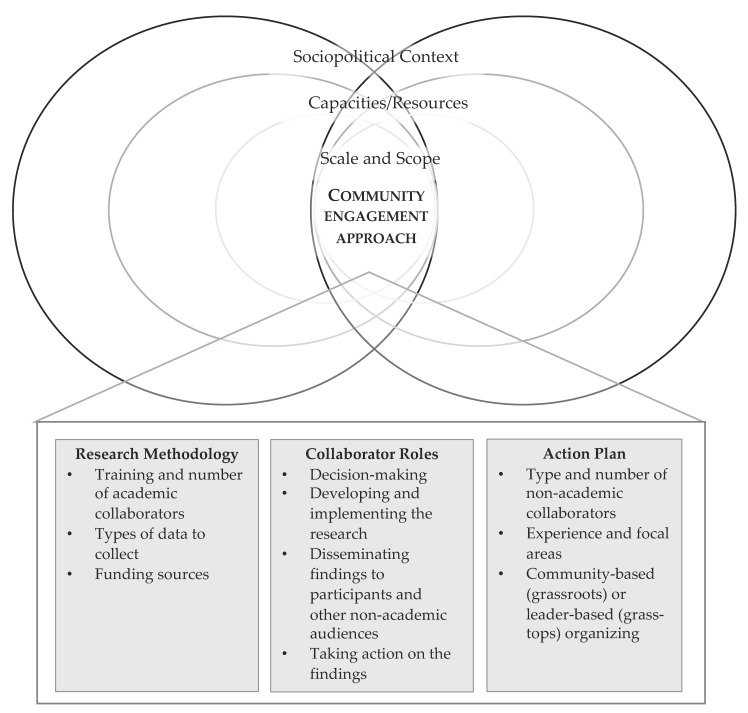
Framework for developing a well-aligned community engagement approach.

**Figure 2 ijerph-17-01187-f002:**
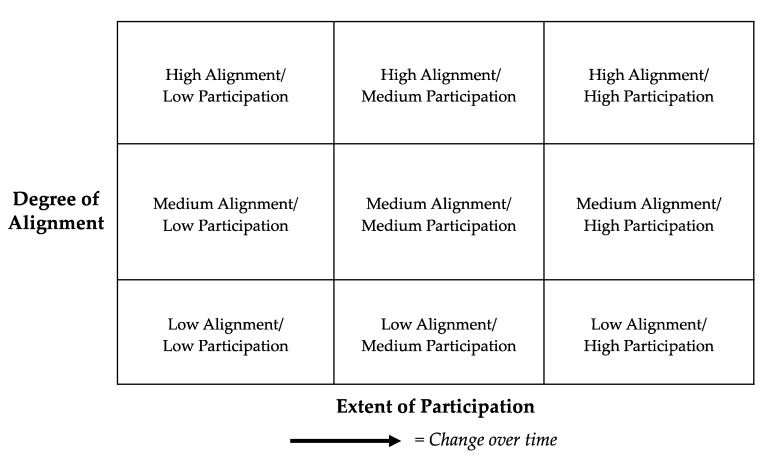
Matrix of community engagement.

**Figure 3 ijerph-17-01187-f003:**
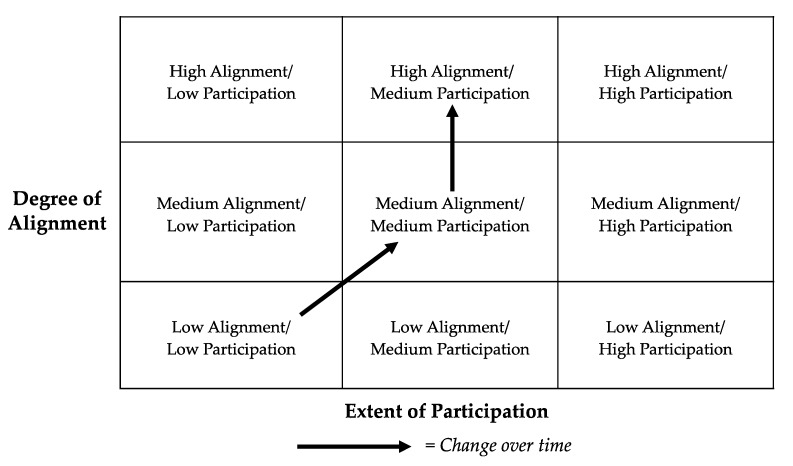
Extent and Alignment of Community Engagement: Harvard University Case Study.

**Figure 4 ijerph-17-01187-f004:**
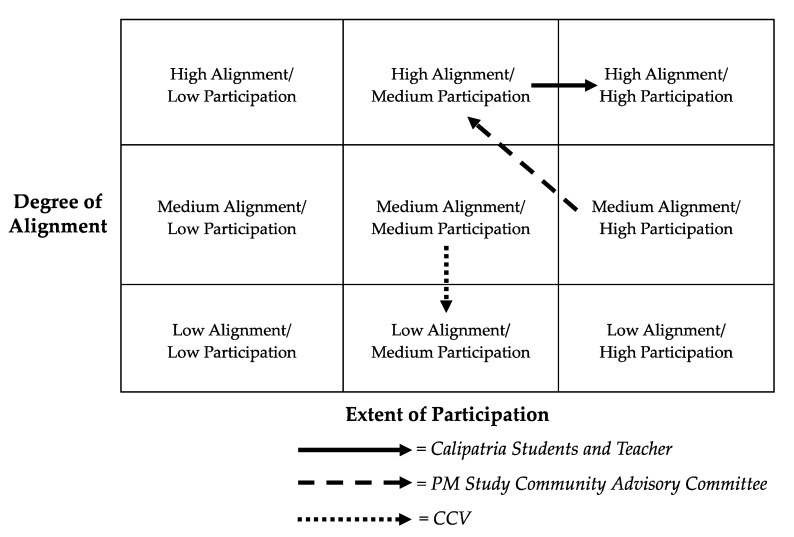
Extent and Alignment of Community Engagement: UC Davis Case Study.

**Figure 5 ijerph-17-01187-f005:**
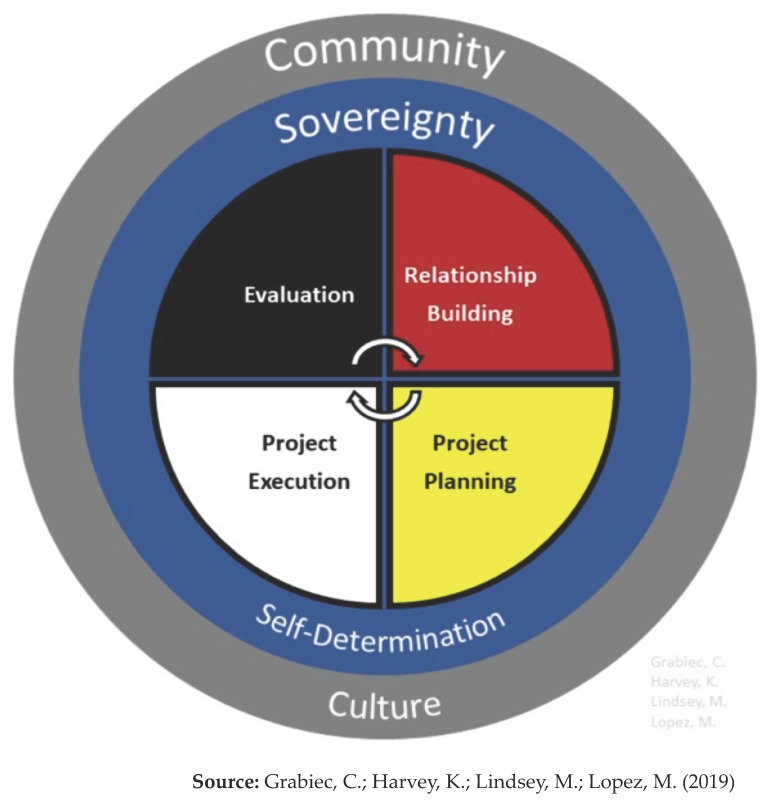
Indigenous Research Framework developed by the University of Arizona SWEHSC CEC.

**Figure 6 ijerph-17-01187-f006:**
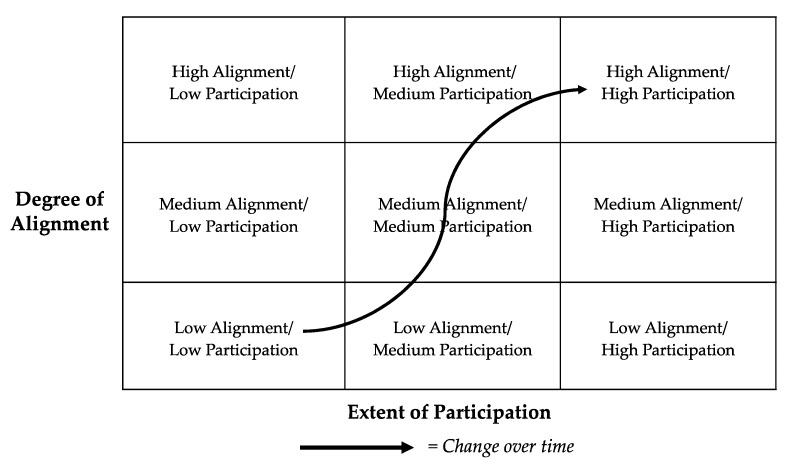
Extent and Alignment of Community Engagement: Arizona Case Study.

**Figure 7 ijerph-17-01187-f007:**
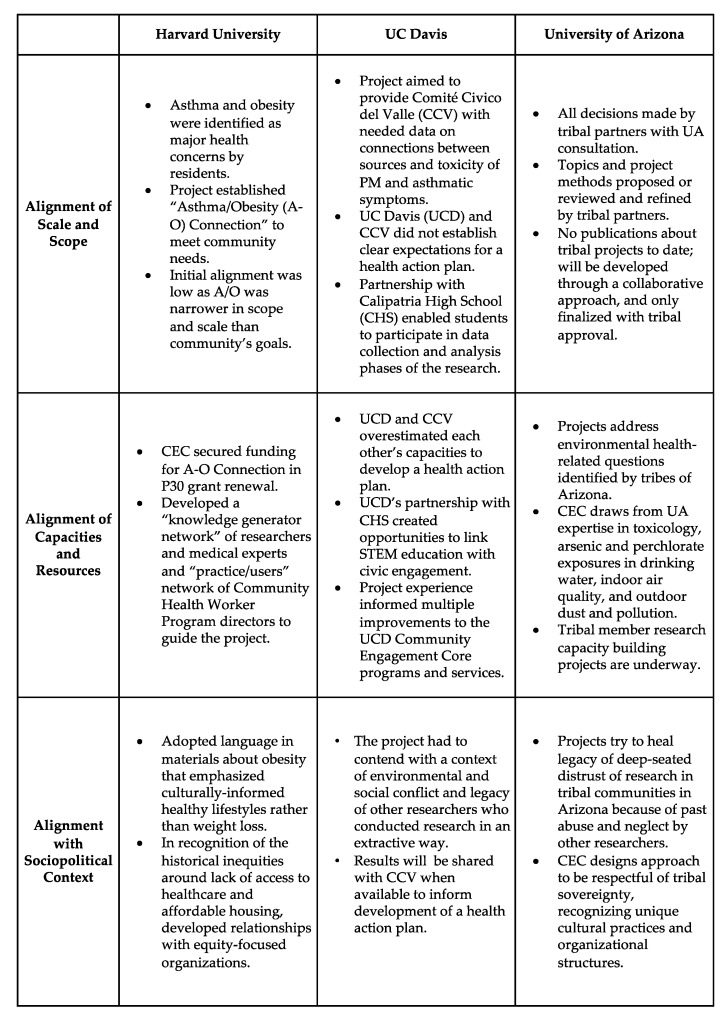
Summary of Alignment in Case Studies.

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
