# Peer review of "Aligning Community-Engaged Research to Context"

_ijerph, 2020, doi:10.3390/ijerph17041187_

Round 1

Reviewer 1 Report

The judgment and interpretation of the relationship between the extent of participation and degree of alignment as shown in the matrix of community engagement (low, medium, high) in three cases do not inform fully. It would be better to indicate the criteria or principles of assessment and explain how to manipulate. In the framework for developing a well-aligned community engagement approach, the Research Methodology, Collaborator Roles, Implementation Plan of three cases need to be illustrated briefly.

Author Response

The authors thank the reviewer for these excellent and helpful comments. We have made the following revisions to address these comments.

We have corrected the affiliation of the authors and made a large number of punctuation/grammar errors. We have added the terms university partner and community partner to Figure 1. We have reduced the length of each case study (and cut 950 words from the overall manuscript), with a focus on the context sections. However, the overall length of the cases has not changed greatly as we have added case study summary tables at the request of another reviewer.

Reviewer 2 Report

Dear Authors,

The presented manuscript provides valuable information on how to effectively engage communities in research. I believe the framework that was provided and its explanation via case studies was very insightful. However, I do have a few comments for you to consider.

The academic affiliation and email address for Marti Lindsey is missing from the title page. The paper itself is very long. I would suggest the you review the context section for each case study to determine what information is absolutely necessary for readers to know and understand the framework/concepts that you have presented in your paper. I believe a more concise summary would suffice. There a are couple areas where I found minor punctuation/grammar error. For example, page 3 line 95 there is a period within parentheses where, in this case, it should outside of the parentheses.  For Figure 1 it would be helpful to the reader to label the larger circles "University Partner" and "Community Partner". The reader does not know what those particular circle represents until they read the text. The figure should be somewhat self-explanatory in that respect. 

Author Response

The authors thank the reviewer for these excellent and helpful comments. We have made the following revisions to address these comments.

We have added improved explanations of the matrix of community engagement (low, medium, high) in each case study throughout the narrative. We have added references to the project methodology in each case study to address the reviewer’s critique of the methods component of the paper. We have created summary tables that address the Research Methodology, Collaborator Roles, Implementation Plans in terms of scale and scope of goals, partnership capacities and resources, and alignment with sociopolitical environment. We have indicated the criteria for assessment of each case and highlighted how the cases could have been improved (or “manipulated” in the terms of the reviewer).

Round 2

Reviewer 1 Report

  The authors have made the relevant revisions to respond to my comments, the article could be accepted after minor revision (eg. to present the Fg. 1 in a clearer way).